# Shared and distinct microRNA profiles between HT22, N2A and SH-SY5Y cell lines and primary mouse hippocampal neurons

Ronan Murphy[1], Javier Villegas-Salmerón[2,3,4], Amaya Sanz-Rodriguez[2,3], Tobias Engel[2,3], Catherine Mooney [3,5], David C. Henshall[2,3]*, Eva M. Jimenez-Mateos [1]*

1 Discipline of Physiology, School of Medicine, Trinity College Dublin, The University of Dublin, Dublin, Ireland, 2 Department of Physiology and Medical Physics, RCSI University of Medicine and Health Sciences, Dublin, Ireland, 3 FutureNeuro Research Ireland Centre, RCSI University of Medicine and Health Sciences, Dublin, Ireland, 4 The SFI Centre for Research Training in Genomics Data Science, RCSI University of Medicine & Health Sciences, Dublin, Ireland, 5 School of Computer Science, University College Dublin, Dublin, Ireland

* dhenshall@rcsi.ie (DCH); jimeneze@tcd.ie (EMJ-M)

## Abstract

MicroRNAs (miRNA) are small non-coding RNAs that are key negative regulators of gene expression. Their roles include shaping the gene expression landscape during and after brain development by defining and maintaining levels of proteins that generate the distinct morphological and functional properties of neurons and other brain cell types. HT22, N2A, and SH-SY5Y are common immortalized neuronal cell lines that offer simple, less expensive, and time-saving options for *in vitro* modelling to evaluate miRNA functions. The extent to which these lines reflect primary neurons remains, however, unclear. Here, we benchmarked the miRNA profiles of cultured mouse hippocampal neurons against Argonaute-loaded miRNAs in the adult mouse hippocampus and miRNA data from the hippocampus of control human donors. We then compared the miRNA expression landscape in HT22, N2A and SH-SY5Y against mouse hippocampal primary cell cultures. We profiled over 700 miRNAs across the lines and detected 310 miRNAs in all four cell types. This included detection of neuron-enriched miRNAs such as miR-124 and miR-128, although the cell lines typically displayed lower levels of these than in primary neurons and reference adult hippocampal tissue. The miRNA profile in the HT22 cell line showed the highest correlation to the mouse primary neuronal cultures. Together, this study provides a dataset on basal miRNA expression across commonly used cell lines for neuroscience research and evidence for both conserved and distinct profiles that should be used to inform decisions on cell lines for modelling brain and miRNA research.

**Data availability statement:** All relevant data are within the manuscript.

**Funding:** This publication has emanated from research conducted with the financial support of the European Union's 'Seventh Framework' Programme (FP7) under Grant Agreement no. 602130. Additional support is from Health Research Board under Grant number ILP-POR-2022-029, Wellcome Trust-IISF call under Grant number Ph2IISF-16585, and Taighde Éireann – Research Ireland, under Grant number 17/CDA/4708, 22/FFP-P/11333 and 21/RC/10294_P2 at FutureNeuro Research Ireland Centre for Translational Brain Science. The funders had no role in study design, data collection and analysis, decision to publish, or preparation of the manuscript.

**Competing interests:** The authors have declared that no competing interests exist.

## Introduction

MicroRNAs (miRNA) are small non-coding RNAs (22–24 nucleotides) that post-transcriptionally regulate gene expression. This is achieved by the miRNA being loaded into the binding pocket of an Argonaute (Ago) protein, thereby forming an RNA-induced silencing complex (RISC) that recognizes complementary sequences in the 3' untranslated region of the targeted messenger RNA (mRNA) [1]. MiRNAs are master regulators of gene expression, with individual miRNAs able to regulate the translation of up to hundreds of protein-coding mRNAs. The impact of a single miRNA on the protein level of a target is, however, generally modest, suggesting that cells use miRNAs to fine-tune their protein landscape [2]. MiRNAs are essential for organism development and are key regulators of cellular identity, with disruption of miRNA production resulting in embryonic lethality in worms, flies, zebrafish, and mice, mainly due to neuronal defects [2,3]. In the adult brain, miRNAs remain essential for homeostasis, with disruption of miRNA production leading to neurological and neurodegenerative conditions [4,5], including epilepsy [6]. In the brain, miRNAs show distinct expression patterns specific to developmental stages, brain regions, and cell types [7–10], reflecting their critical role in neuronal differentiation and establishing and maintaining neuronal diversity [11,12]. For example, miR-124 is key for establishing the neuronal lineage during embryogenesis via regulation of the transcription factor REST and suppressing non-neuronal transcripts in neurons [13–15]. MiR-125 is critical for neuronal maturation and maintaining neuronal properties [16], and miR-133b is expressed in and regulates the maturation of dopaminergic neurons in the midbrain [17]. Among key processes shaped by the actions of miRNAs are neuronal structure and function, including dendritic morphology and the complement of neurotransmitter receptors, ion channels, and transporters [2]. Consequently, miRNAs have been extensively studied as potential mediators of aberrant gene expression in neurological diseases [2,18].

Immortalized cell lines are commonly used to understand and identify molecular and cellular processes. In classical studies, HeLa cells were used to demonstrate that miR-124 is critical to establishing the neuronal gene expression landscape [19]. Cell lines are an invaluable tool that can provide a simple way to identify and test molecular and cellular mechanisms before moving to primary cultures or in vivo experiments [20–23]. However, immortalized cell lines typically have a tumoral origin or have been modified to be kept cultured indefinitely, which results in phenotypes and patterns of gene regulation that do not reflect some neuronal characteristics and responses to stimuli. Examples of common immortalized neuronal cell lines are the HT22, N2A and SH-SY5Y cells. HT22 cells are immortalized primary mouse hippocampal neuronal cells, derived from parent HT4 cells [22,24]. Undifferentiated HT22 cells express low levels of cholinergic markers and glutamate receptors [25,26], in contrast to mature hippocampal neurons. Differentiated HT22 cells express higher levels of N-methyl-d-aspartate receptor (NMDAR) mRNA [27], making them susceptible to glutamate-induced excitotoxicity [26]. N2A is a mouse neuroblastoma cell line, commonly used to study neuronal differentiation and used extensively to test novel cancer treatments [28]. SH-SY5Y are neuroblastoma cells of human origin with a neuroblast-like morphology and expression of immature neuronal markers [29]. After

differentiation with retinoic acid and neurotrophins, SH-SY5Y cells become morphologically similar to primary neurons and express neuron-specific markers [29].

Previous studies have evaluated the expression of miRNAs in cortical primary neurons and glial cells (astrocytes, microglia and oligodendrocytes) [8]. This identified a set of miRNAs that are enriched in neurons, with expression fivefold higher than the glial cells, including miR-124, miR-132 and miR-135b, while the same cells display very low expression of miR-21 and miR-146a compared to glial cells [8]. Importantly, neurological disorders can result in altered miRNA expression, contributing to pathology [30]. Here, cell lines have contributed to discoveries on miRNAs, including linking miR-134 to the pathophysiology of epilepsy [31–37].

While the miRNA profiles of neuronal cell lines have been described, researchers typically report on individual cell lines. Having a catalogue of miRNA profiles of multiple cell lines analysed together is important for the selection of the most appropriate cell line for a particular research project. Here, we established the basal miRNA profile for three commonly used immortalized cell lines in neuroscience research and compared these to the profile in primary hippocampal neurons from mice and brain data. The data will support decision-making on the best cell line to study specific miRNAs or molecular mechanisms before moving to *in vivo* experiments.

## Materials and methods

### Primary cell culture

All procedures were performed in accordance with the guidelines of the European Communities Council Directives (86/609/EU and 2010/63/EU) and were reviewed and approved by the Research Ethics Committee of the Royal College of Surgeons in Ireland (REC #765) under license from the Department of Health, Dublin, Ireland (B100/4524).

Primary cultures of hippocampal neurons were prepared following a previously described protocol [34]. C57Bl/6J pregnant females were obtained from the Biomedical Research Facility. Embryonic day 18 (E18) mouse brains were removed and placed into a solution of HBSS (Sigma-Aldrich Ireland Ltd., Wicklow, Ireland). Hippocampi were dissected and cells were disaggregated using papain (100 units/ml, Sigma-Aldrich Ltd., Wicklow, Ireland) for 30 min at 37 °C. Neurons were then isolated by tissue dissociation and plated onto a 1 mg/ml poly-L-lysine and 20 µg/ml laminin. Cells were maintained in Neurobasal medium supplemented with B-27 and N2 (Thermo-Fisher, MA, USA) at 37 °C in a humidified atmosphere with 5% (v/v) CO2 for 10 days. Half of the medium was changed to fresh medium every second day. RNA was isolated at day *in vitro* 10 (DIV10) as described below.

### Cultured cell lines

HT22 cells (a gift from Prof C. Culmsee) were grown in Dulbecco's Modified Eagle Medium (DMEM, Lonza Biologics, UK), 4.5 g/L high glucose with glutamine and sodium pyruvate containing 10% FBS (fetal bovine serum). SH-SY5Y cells (a gift from Prof J.H.M Prehn) were cultured in Dulbecco's Modified Eagle's Medium-F12. N2A cells (a gift from Prof J.H.M Prehn) were cultured in a supplemented DMEM medium (ThermoFisher, MA, USA). All immortalized cell lines were supplemented with penicillin/streptomycin, and incubated in 5% $CO_2$ humidified atmosphere at 37°C. The immortalized cell lines were plated at a 10.000 cells/mm$^2$ density in an M24 plate and harvested 48 hours later. For triplicates, the vial was defrosted in three independent plates and maintained as three independent cell cultures. After four passages, cell lines were seeded in an M24 plate, and RNA was extracted as described below. Cell lines were tested for mycoplasmas before experiments started and results were negative for the three immortalized cell lines.

### RNA extraction and OpenArray

Total mRNA was extracted using the Trizol protocol as previously described [34,38]. Quantity of mRNA was measured using a Nanodrop Spectrophotometer (Thermo Fisher Scientific, Rockford, IL, USA), and RNA dilutions were made up in

nuclease-free water. 100ng of purified RNA was processed by reverse transcriptase and pre-amplification steps following the manufacturer's protocol (Applied Biosystems). The pre-amplification reaction was mixed with TaqMan OpenArray Real-Time PCR Master mix (1:1). The mix was loaded onto the OpenArray rodent microRNA panel (750 mouse/rat miRNAs (Cat number: 4461105)) and ran using a QuantStudio 12K Flex PCR (Life Technologies).

## Analysis of OpenArray

Analysis and statistical analysis of the OpenArray miRNA profiling data were performed in R. Data were first filtered according to amplification score (AmpScore >= 1.24), quantification cycle confidence (Cqconf >= 0.8) and cycle threshold (Ct > 10 and < 35). For the comparison of the expression across the four cell types, miRNAs were included if they were present in at least 2 of the 3 samples in each group. Missing data points were imputed as the mean Ct of the other two samples, and the data were quantile normalised using the Bioconductor package qpcrNorm [39]. Principal component analysis (PCA) was performed on the quantile normalised data. Heatmaps and Venn diagrams were generated using the Complex Heatmap [40] and Eulerr [41] packages in R, respectively. Correlation was calculated using the Pearson correlation coefficient and the Mean Squared Error to evaluate similarity. Correlation graphs, slope graphs and PCA plots were generated using ggplot2 [42]. Differential expression analysis was performed using the limma package [43]. p-values were adjusted for multiple testing by controlling the false discovery rate (FDR) according to the method of Benjamini and Hochberg [44]. A miRNA was considered to be differentially expressed if the adjusted p-value was < 0.05. Relative expression was analysed using the ΔΔCt method with all miRNA analysis normalised by subtracting the geometric mean Ct for each miRNA. Statistical analysis on individual miRNAs was performed using ANOVA followed by Dunnet test. Significance was assigned as p < 0.05. Raw data is available in the Supporting Information (Open array Cells file).

Further analysis was carried out to evaluate if the distribution of the number of miRNAs expressed in each condition is due to chance using the Bootstrapping resampling method. Bootstrapping iterations were performed using random binomial resampling with probability equal to the percentage of miRNAs expressed by each cell type in the original data (see S1 Fig and S1 Methods).

## Human data and RNA-seq analysis

RNA-seq raw data of hippocampus of control human donor was obtained from previously available data [46]. Hippocampal post-mortem tissue was obtained during autopsy resection and a history of no-seizures or other neurological diseases was confirmed as described in the original publication [46]. Analysis of RNA-seq data was performed using the Bioconductor package edgeR [45]. Data were filtered by miRNA counts per sample (count > 10), and only samples with ~ 8 million reads were used for the analysis. Only miRNAs present in all 3 samples were included in the analysis. Data were standardised by log counts per million followed by a linear transformation to convert the count data to the equivalent Ct range (10–35). Next, the data were merged with the qPCR data and quantile normalised using the Bioconductor package qpcrNorm [39]. Differential expression analysis was performed as described above.

## Results

### Cultured mouse primary hippocampal neurons have a similar miRNA profile to Ago2-bound miRNAs from the adult mouse hippocampus

Mouse primary hippocampal neurons were selected as a suitable comparator for the miRNA profiles of HT22, N2A and SH-SY5Y cells (Table 1). First, and using previously published data [38,46], we compared our in vitro hippocampal primary neurons from mice (DIV (Day in vitro) 10) to the miRNA profile of the mouse hippocampus by comparing with Ago-2 bound miRNAs from the hippocampus of adult C57BL/6 mice [38]. We also compared the profile to miRNAs in the adult human hippocampus [46]. When compared with mouse tissue, we found that 78% of the expressed miRNAs in our

**Table 1. Characteristics of primary hippocampal neurons and main cell lines used in the current manuscript. In the current study, 1e6 hippocampal neurons were cultured for 10 Days. Immortalized cell lines were used 2 days after plated, and at a maximum confluence of 80%.**

| | *Primary hippocampal neurons* | *HT22* | *N2A* | *SHSY5Y* |
|---|---|---|---|---|
| *Species* | Mus musculus-C57BL6 | Mus musculus | Mus musculus-Albino Strain A | Homo sapiens- 4 years old female |
| *Cell origin* | Embryonic Day 18- Hippocampus | HT4 derived cell- Immortalized with SV 40T virus (hippocampus) | Spontaneous tumour | Bone marrow (patient with a diagnosis of neuroblastoma) |
| *Immortalized* | No | Yes | Yes | Yes |
| *Main Applications* | Ex vivo neuronal function | Glutamatergic neurotoxicity | Neuronal differentiation | Adrenergic cells |

primary mouse neurons were also among those actively loaded in the RISC in the mouse brain (Fig 1A). When compared with human hippocampus, 91% of the expressed miRNAs in our primary mouse neurons were also detected in the brain tissue (Fig 1B). When we compared all three groups, 119 miRNAs were commonly detected, including various neuronal expressed miRNAs (Fig 1C) such as miR-128-, miR-132, miR-134, miR-135a, miR-181a/c, miR-212 and miR-218. Of note, only 8 miRNAs were exclusively detected in primary hippocampal neurons (Fig 1C; miR-44b, miR-422a, miR-551b, miR-601, miR-659, miR-661, miR-672 and miR-1244). Then, we focused on the cell expression of the 160 detected miRNAs in primary neurons, 158 miRNAs were identified as highly enriched in neurons, including miR-128, miR-132, miR-134 and miR-218 [9] (Fig 1D, top panel). A higher proportion of non-neuronal miRNA was detected in hippocampal tissue from mice (Fig 1D, middle panel) and human brain tissue (Fig 1D, bottom panel); 3% in both data sets.

## miRNA profiles in primary hippocampal neurons and HT22, N2A, and SH-SY5Y cell lines

Next, we evaluated the extent to which the three immortalised cell lines overlapped with the miRNA expression profile in our primary hippocampal neurons. The miRNA profiling was carried out using the qPCR-based Open Array Platform (Fig 2A). Principal Component Analysis (PCA) using the total miRNA datasets from each cell type showed separation of primary hippocampal neurons, HT22, N2A, and SH-SY5Y into four distinctive groups (Fig 2B). We detected 310 miRNAs in the four cell types (Fig 2C). Of these, 32% (98 miRNAs) were commonly detected in all the cell types. Each cell type expressed a unique subset of miRNAs; 13 miRNAs were detected only in the primary hippocampal neurons, 7 in HT22, 20 in N2A, and 61 in SH-SY5Y (Fig 2C). Then, we compared the primary hippocampal neurons with each cell line individually. The SH-SY5Y and N2A cell lines displayed the highest overlap with primary neurons, with 10 and 9 miRNAs respectively. HT22 and the primary hippocampal neuron had only 1 miRNA exclusively detected in both cell types (Fig 2C). To determine whether the observed overlap in miRs between the different cell lines was due to random chance, we performed 1000 bootstrapping iterations and compared the median results to our original data (S1 Fig and S1 Methods). This revealed that the resampled distribution was significantly different from the distribution in our profile (S1 Fig), and thus the overlap in miR expression across the cell lines is likely due to biological similarity rather than random chance.

When we evaluated the level of miRNA expression in all cell types (Fig 2D), miRNAs such as miR-16, miR-30c, miR-125a have similar expression levels in the four cell types (Fig 2D). In contrast, miR-9, miR-132, miR-138 and miR-218 were highly expressed in neurons compared to the immortalized cell lines (Figs 2D and S2).

## Neuron-enriched miRNAs in primary neurons and cell lines

Next, we explored the identities and relative expression of the 98 commonly detected miRNAs across the cell types (Fig 3A and 3B). Primary hippocampal neurons showed the highest expression of known neuron-enriched miRNAs compared to the immortalized cell lines, including miR-128, miR-132, miR-212 and miR-218 (Fig 3B). In contrast, primary neurons

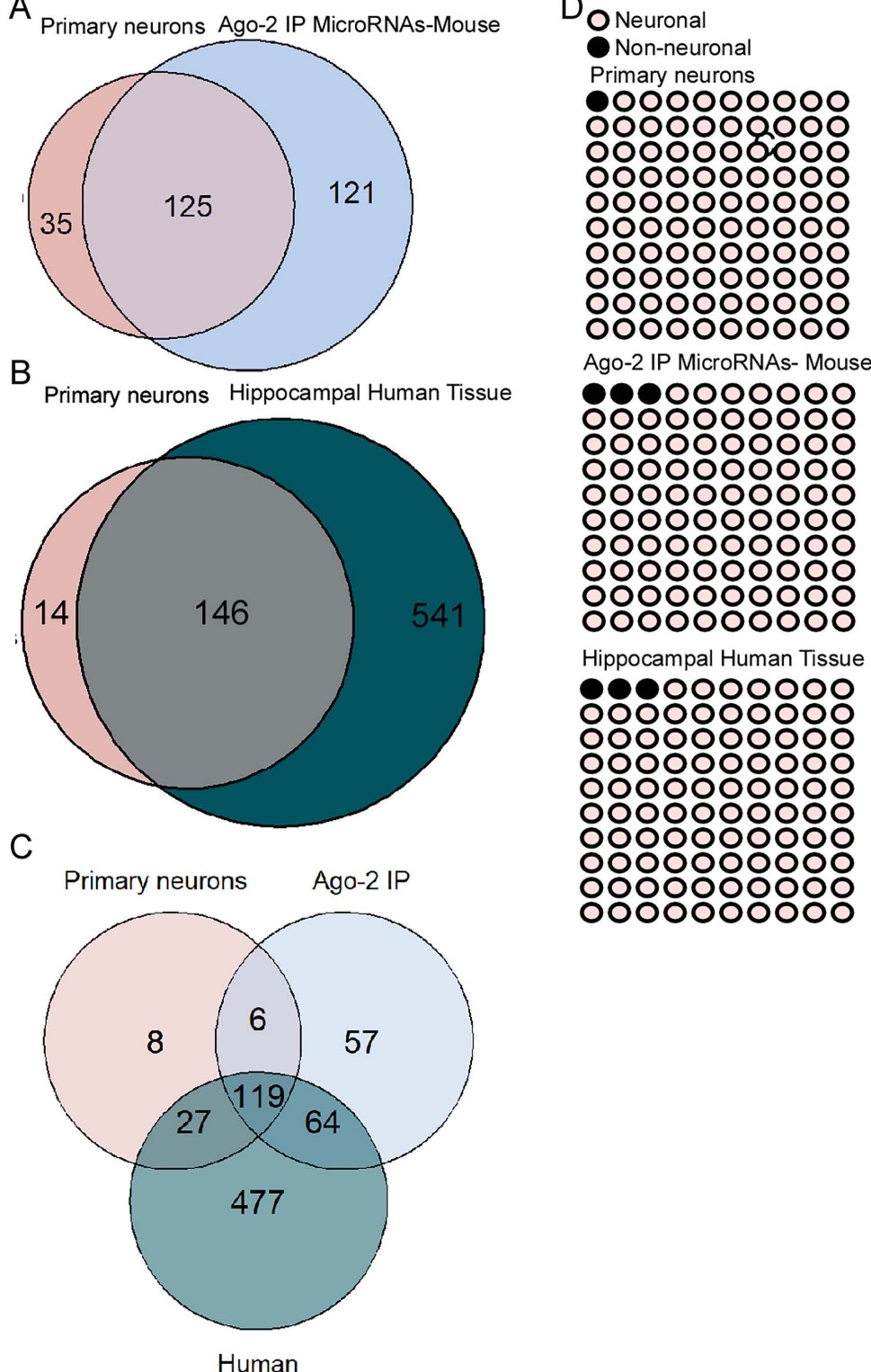

**Fig 1. Primary hippocampal neurons have a similar miRNA profile than adult hippocampus.** A,B,C) Venn diagrams show the overlap on the number of miRNAs between mouse hippocampal primary neurons, and Ago-2 bound miRNAs from the adult mouse hippocampus **(A)**, between human control hippocampal neurons **(B)** and between the three tissue types **(C)**. **D)** Dot plots (10 x 10) show the proportion of neuronal enriched miRNAs in primary hippocampal neurons (top), hippocampi from adult mouse (middle), and human hippocampus (bottom).

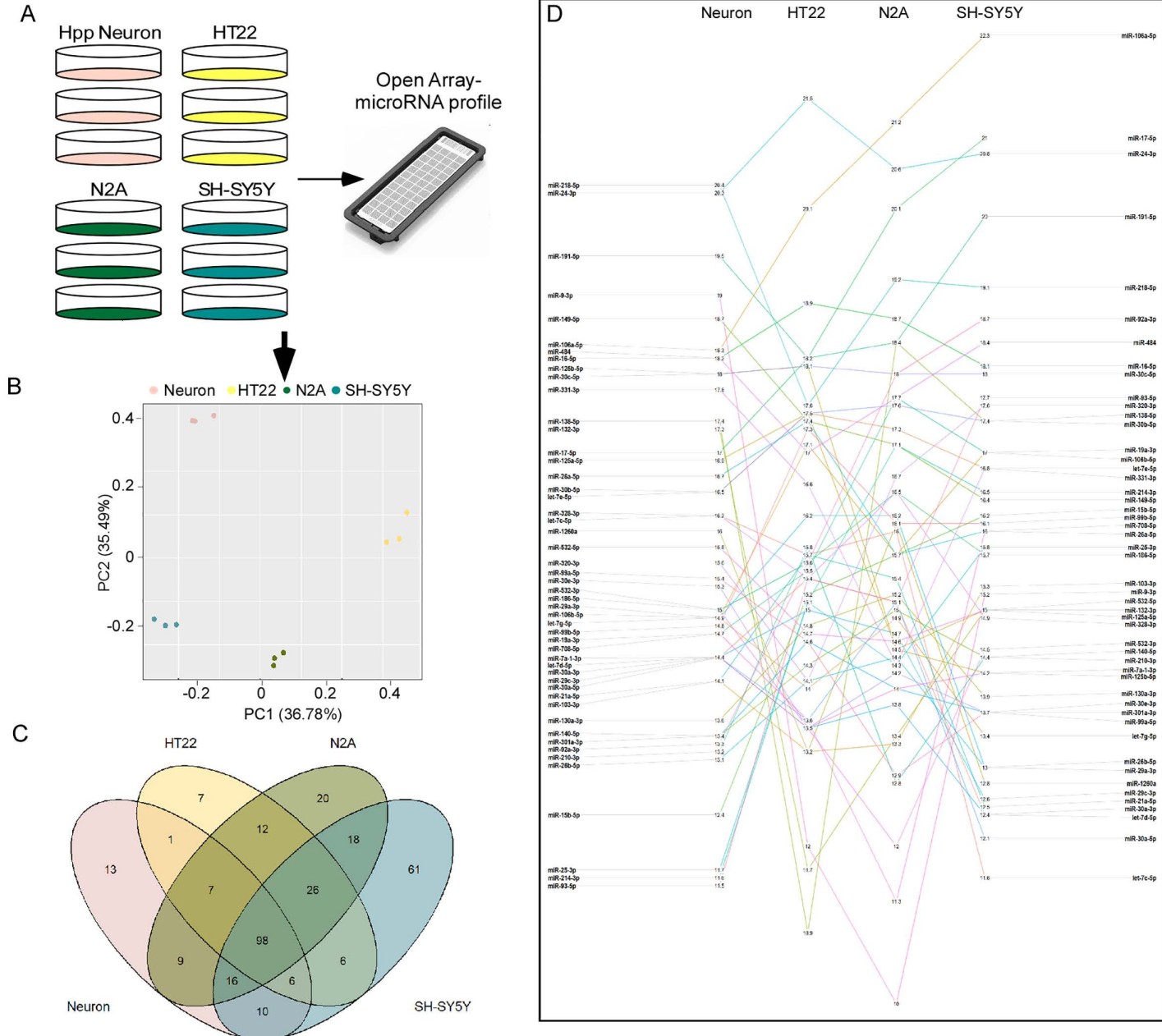

**Fig 2. MiRNA profile in primary hippocampal neurons, HT22, N2A and SH-SY5Y cell lines.** A,B) Schematic of the procedures carried out in carrying study **(A)**, and Principal Component Analysis (PCA, **B**) represent the clusters of the four cell lines. **C)** Volcano Plot represents overlap between the four cell lines. **D)** Slope graph illustrating the relative expression of the 50 most abundant miRNAs present in all four cell types. Relative expression is defined as 35 – mean Ct.

displayed low levels of miRNAs involved in cell proliferation, including the miR-17/92 cluster (miR-27, miR-18 and miR-92) [47] (Fig 2B). The primary neurons also displayed high expression of miRNAs linked to neurological disorders such as epilepsy, including miR-129a, miR-134, miR-135a and miR-146a [30]. In contrast, miR-129 was not detected in HT22 or

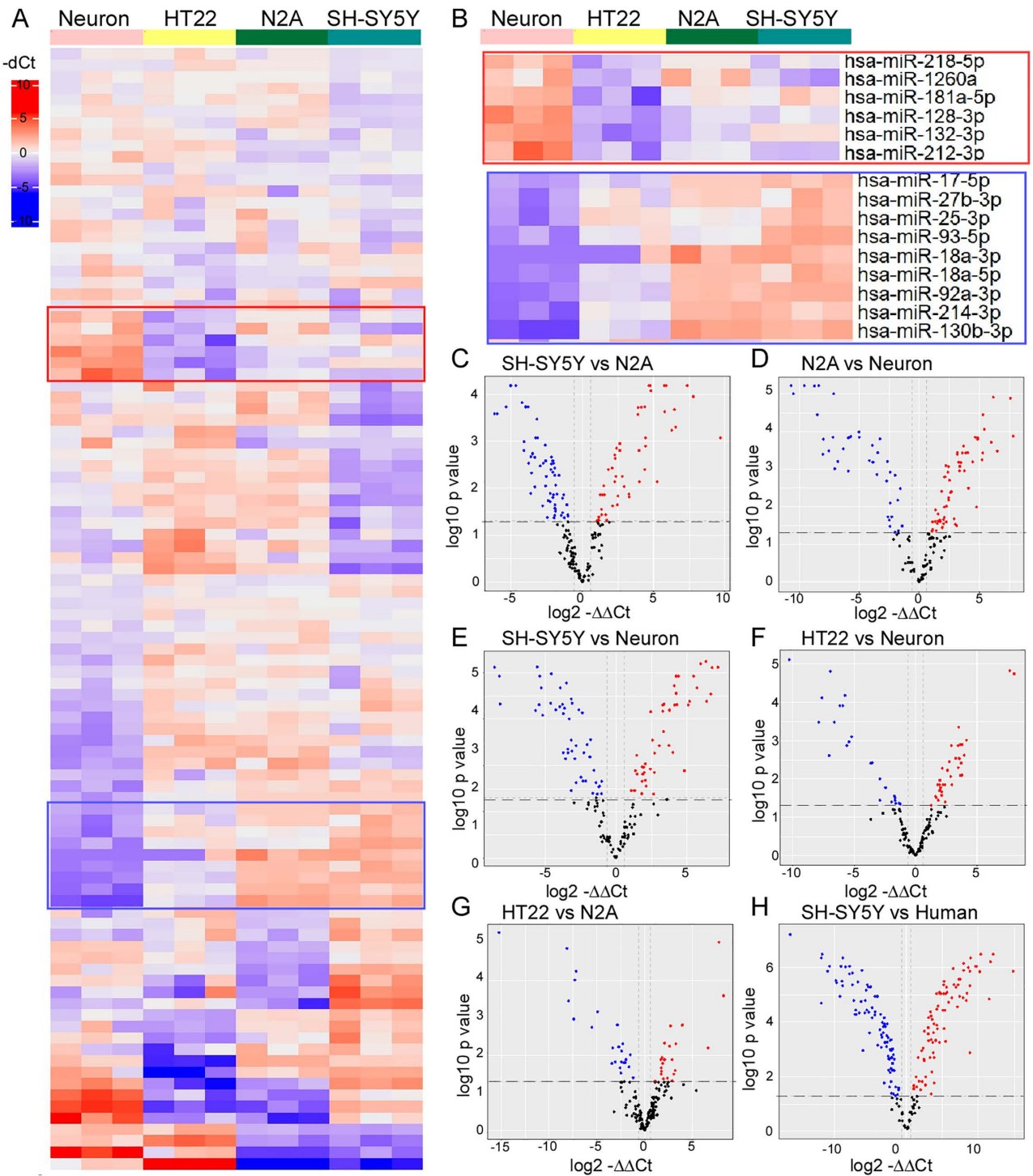

**Fig 3. Comparison of gene expression between the cell lines. A)** Heat map of the 98 miRNAs expressed in all the four cell types, each line represents an independent sample. Red: high expressed, purple: low expression. **B)** Amplified Heat map of the selected miRNAs highly expressed in primary neurons (top) or low expressed (bottom). C-H) Volcano plot shows the log2 FC (Fold changes) indicated by the mean expression level for each gene and each dot represents one gene. **C)** SH-SY5Y versus N2A, **D)** N2A versus hippocampal neurons, **E)** SH-SY5Y versus hippocampal neurons, **F)** HT22 versus hippocampal neurons, **G)** HT22 versus N2A and **H)** SH-SY5Y versus human brain sample.

N2A, miR-134 was not detected in HT22, miR-135a was not detected in N2A, and miR-146a had the lowest expression in HT22 or N2A (Fig 3B).

## Differential expression of miRNAs in cell lines compared to primary hippocampal neurons

To further characterize the basal miRNA profile between the four cell types we performed differential gene expression analyses. Comparing first the two neuroblast cell lines (N2A and SH-SY5Y), we found 200 miRNAs detected with 59 miRNAs with lower expression and 50 miRNAs with higher expression levels in N2A compared to SH-SY5Y (Fig 3C). The number of commonly detected miRNAs between primary hippocampal neurons and both neuroblast cell lines was similar (163 and 166 miRNAs in N2A and SH-SY5Y respectively; Fig 3D and 3E). We detected 142 miRNAs commonly detected between HT22 and primary hippocampal neurons (Fig 3F). From these commonly detected miRNAs, HT22 showed the highest level of similarly-expressed miRNAs, with 57% of the miRNAs expressed at a similar level, followed by SH-SY5Y cells, with 43% of miRNAs expressed at a similar level, and 38% of miRNAs expressed at a similar level between N2A and primary hippocampal neurons (Fig 3D–3F).

To further understand if these similarities are driven by species or cell type, we compared both murine immortalized cell lines, HT22 and N2A (Fig 3G) and the human cell line (SH-SY5Y) to previously published human data [48] (Fig 3H). This revealed 72% of the detected miRNAs were commonly expressed between HT22 and N2A immortalized cell lines (Fig 3G) while 86% of miRNAs were commonly detected between the SH-SY5Y cell line and human tissue (Fig 3H).

## High correlation between miRNA profile in HT22 cells and primary mouse hippocampal neurons

To extend these insights, we evaluated the correlation of the full miRNA profiles between primary neuronal cell cultures and the three immortalized cell lines. This determined that HT22 cells had the highest correlation level (r = 0.735, MSE = 8.515. Fig 4A), while N2A had the lowest correlation level (r = 0.564, MSE = 13.444. Fig 4B), with the SH-SY5Y cell profile falling in an intermediate range (r = 0.715, MSE = 8.443. Fig 4C). We further explored some of the contributors to these correlations (Fig 4). When comparing HT22 cells to primary neuronal cells, miRNAs involved in neuronal function (e.g. miR-9, miR-128, miR-132, miR-135a, miR-135b) and inflammation (miR-146a, miR-212) had lower expression levels (Fig 4A), while miRNAs involved in development and cell division (for example miR-18, miR-27a/b, miR-31, miR-106a/b and miR-222) had higher expression levels in HT22 than in primary neuronal cells cultures (Fig 4A). Similar results were obtained for N2A cells with miR-128, miR-132 or miR-134 displaying lower expression levels in N2A (Fig 4B). Interestingly, let-7e, miR-151, miR-335 or miR-378, which are involved in neuronal differentiation, have higher expression levels in N2A compared to primary cell cultures (Fig 4B).

We next compared the human neuroblastoma SH-SY5Y to the primary neuronal cell line. Here we found miRNAs expressed in adult brain and involved in neuronal function such as miR-124, miR-134 or miR-135a/b also had a lower expression level in SH-SY5Y compared to the primary cell line (Fig 4C) whereas miR-27a, miR-103 or miR-140 had a higher expression level in SH-SY5Y compared to primary cell lines.

Then, we evaluated the biological pathways enriched in all cell lines combined, primary cell cultures and/or each cell line. Axon guidance, cell-to-cell communication, and transport of molecules were the most significantly regulated within the 98 common miRNAs expressed in all the cell types (S3 Fig). Metabolism was the main pathway regulated by miRNAs expressed in primary hippocampal neurons and HT22. Axon guidance and developmental biology was found between primary hippocampal neurons and N2A. And at last, RNA regulation and post-transcriptional regulation, between primary hippocampal neurons and SH-SY5Y cells (S3 Fig). Finally, we evaluated the miRNAs-regulated pathways by the specific miRNAs expressed exclusively in each cell line. This revealed that pathways involved in Calcium regulation and CREB transcription factor signalling activation are only seen in the primary cell culture, while mTOR and S6K signalling pathways are only seen in HT22 cell line, extracellular matrix regulation is observed only in N2A cell lines, and, finally, in SH-SY5Y cell lines has pathways related to cytokine regulation and innate immune system (S4 Fig).

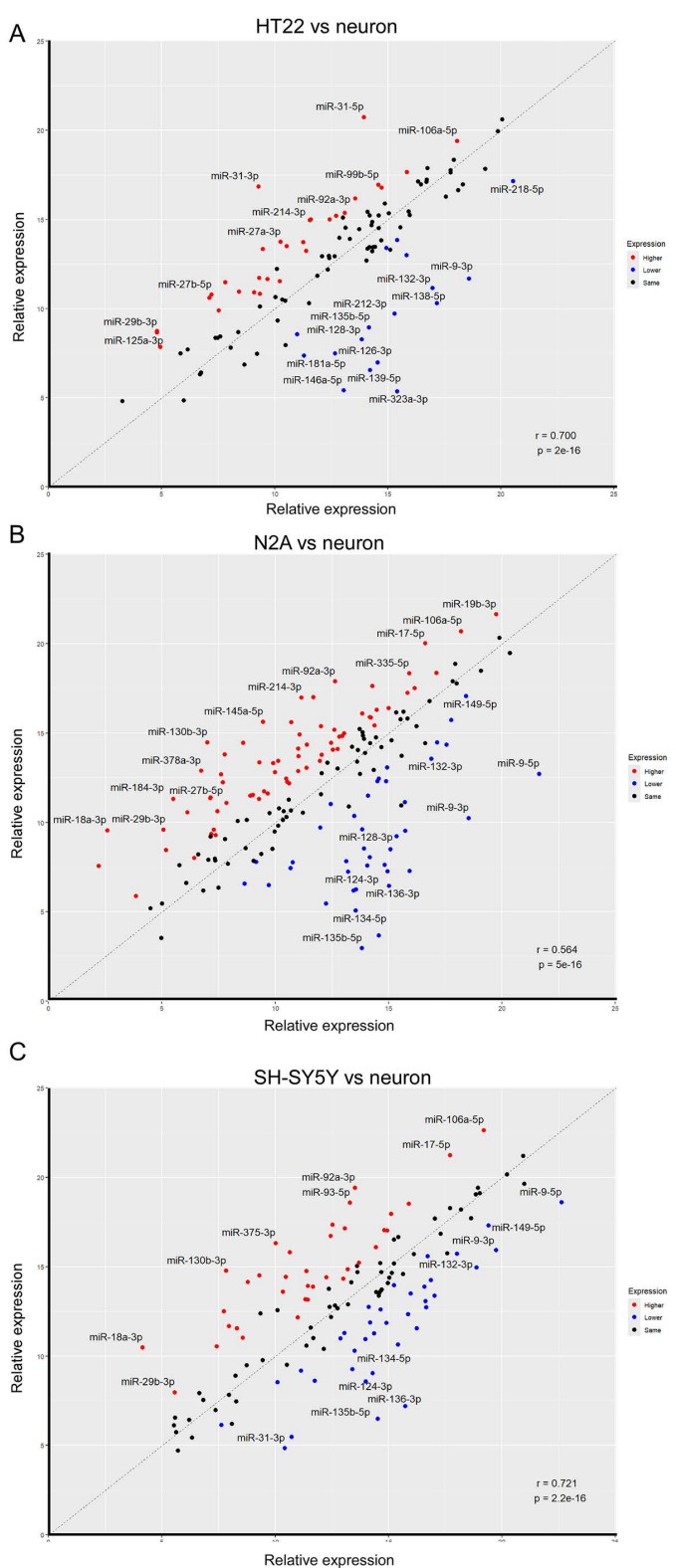

**Fig 4. Correlations between the expression levels of the individual miRNAs between the primary hippocampal neurons and A) HT22 (r=0.700, MSE=8.515); B) N2A (r=0.564, MSE=13.444); and C) SH-SY5Y (r=0.721, MSE=8.443).**

Finally, we explored the expression of additional miRNAs in each cell line, particularly those miRNAs previously linked to neuronal function. Here, we found several miRNAs have similar levels in all cells, for example, miR-30b and miR-30c (Fig 5A). MiR-125b and miR-29a have similar levels only between HT-22 and neurons (Fig 5B). While miR-106b and miR-28 have similar levels only between N2A and neurons (Fig 5C) and miR-16 and miR-29 have similar expression levels between SH-SY5Y and neurons (Fig 5D). Together, the results represent a comprehensive mapping of the basal miRNA expression profiles in commonly used neuronal cell lines which may be useful to inform choice of cell line in miRNA research on the brain.

## Discussion

Immortalized neuronal cell lines are invaluable tools to test the targets and function of key miRNA as well as test the properties of experimental drugs [49]. However, the suitability of cell lines for neuronal miRNA studies has often neglected an assessment of their basal miRNA expression and its relevance to primary neurons or the brain. Here, we compared three well-used immortalized neuronal cell lines, HT22, N2A and SH-SY5Y to primary mouse hippocampal neurons to evaluate their similarities and differences and support decision-making on the best cell line to study specific miRNAs or molecular mechanisms before moving to primary cell cultures or *in vivo* experiments. Importantly, the current study compared hippocampal primary neurons not only to the HT22 line which are of hippocampal origin, but also neuroblastoma cell lines N2A and SH-SY5Y. This adds important value to the present findings in two ways. First, the inclusion enables the identification of miRNAs signatures that reflect tissue origin versus immortalisation. Second, the inclusion of these cell lines gives the fingerprint of the similarities between both neuroblastoma cell lines and primary hippocampal neurons. Overall, the study supports the general suitability of each cell line as well as identifying profiles which may favour the selection of one cell type over another, depending on the research question. The findings provide a useful resource for experimental planning including assessing the therapeutic potential of miRNAs for neurologic diseases such as epilepsy.

A key starting point in the current study was to compare cultured primary neurons to the miRNAs that are active in the adult mouse brain. We observed that 79% of miRNAs expressed in our mouse primary hippocampal neurons were also present in the RISC complex in the adult mouse hippocampus [38], with similar results found when compared with human tissue [46]. Thus, cultured mouse neurons are a suitable proxy for the adult brain providing a benchmark for the later comparison to the cell line profiles. The small discrepancies between primary neurons and the brain (hippocampus) may reflect maturity differences or suggest a pool of miRNAs is not Ago-2 bound in the cultured neurons, perhaps as they require regulatory signals such as neuronal activity [50]. Differences may also be due to the influence or abundance of glial cells, which may be lower in the primary hippocampal neurons. Our mouse primary hippocampal cell cultures have a high (approximately 80%) shared profile of miRNAs with results from rat primary hippocampal cells [51]. This includes similar expression of miR-125a, miR-125b, miR-184, miR-195, miR-214 and miR-384, which in rat and mouse hippocampal neurons are in the highest expression group. Furthermore, miR-22, miR-24, miR-26, miR-29a, miR-124 and miR-129, are within the lowest expressed miRNAs; and miR-103, miR-130a, miR-130b, miR-134, miR-376 and miR-449 are in the group with intermediate expression [44]. However, this differed from mouse cortical primary neurons, where the highest expressed miRNAs are miR-124, miR-132 and miR-135b [9], suggesting that the brain region may influence miRNA levels. Importantly, a larger number of miRNAs were only detected in the human tissue. This may reflect, however, that RNAseq was used to generate those data which allows for a broader range of detection than the Open Array Platform.

While cell lines cannot fully replicate the characteristics of primary hippocampal neurons, the present study reveals that HT22, N2A and SH-SY5Y cells express miRNAs that make them suitable for the analysis of neuronal and/or brain-related processes. Specifically, each cell line expressed a large number of the same miRNAs as are present in mouse primary hippocampal neurons. For example, N2A cells have similar miR-106b and miR-28 expression levels than primary hippocampal neurons, and SH-SY5Y cells have similar miR-16 and miR-29 expression levels than primary hippocampal neurons. Similar to both neuroblastoma cell lines, miR-125b and miR-29a have similar expression levels between HT22 and

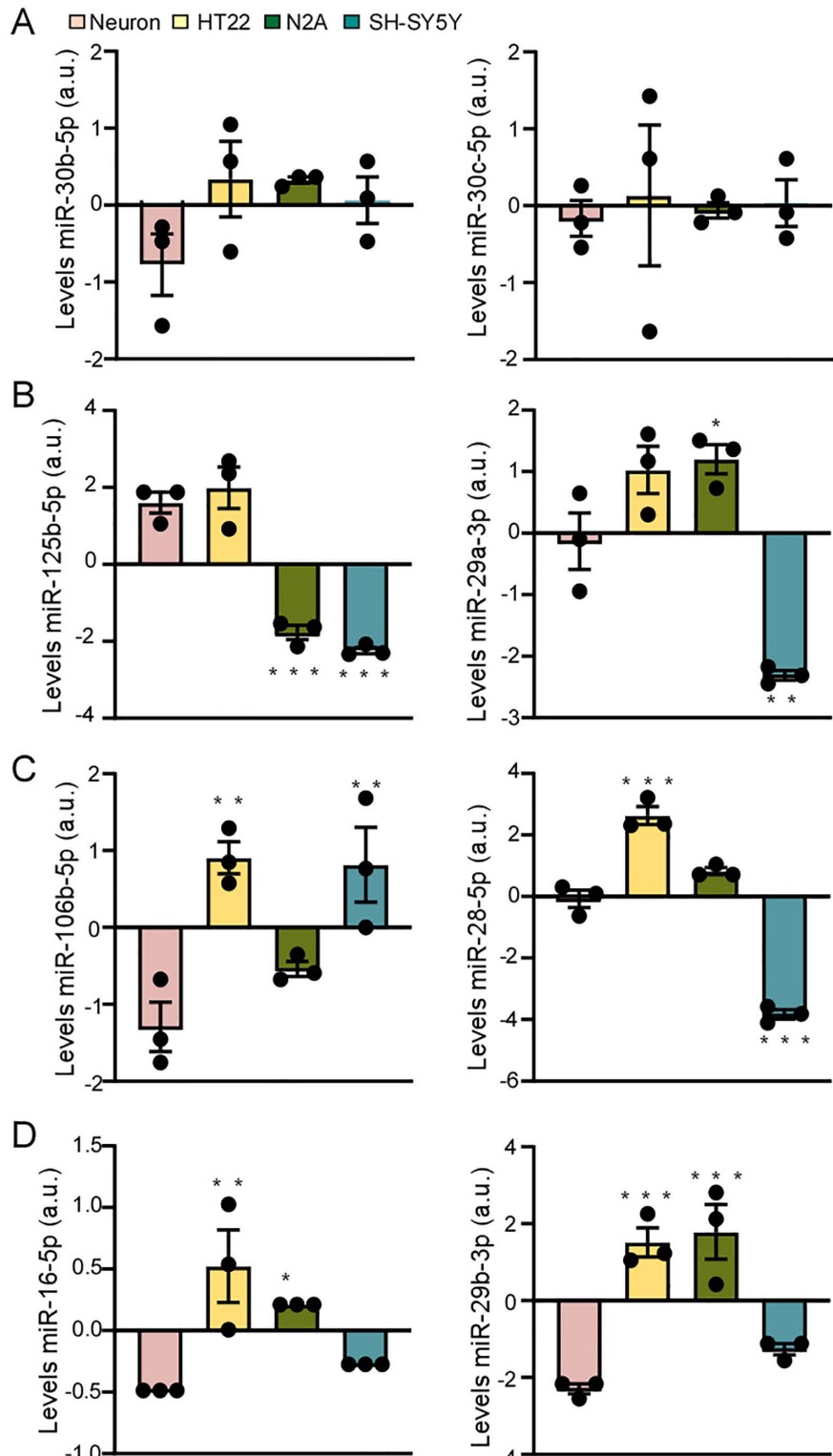

**Fig 5. Individual miRNA levels in the four cell lines, primary hippocampal neurons (pink), HT22 (yellow), N2A (green) and SH-SY5Y (blue). A**) miR-30b and miR-30c. **B**) miR-125b and miR-29a. **C**) miR-106b and miR-26. **D**) miR-16 and miR-29b. *p < .05; **p < 0.01; ***p < 0.001.

primary hippocampal neurons. Since both miRNAs are involved in regulating processes of neuronal death [52–55], HT22 cells may be suitable for analysing neuronal apoptosis regulated by miRNAs.

A key observation in the present study was that while miRNA profiles show substantial overlap with primary mouse neurons, they tend to express lower levels of adult-enriched miRNAs. This is not surprising and suggests researchers should be wary that the cell lines are a better model for neurodevelopment and may retain tumor-related miRNA profiles. We note, however, that the cells were not differentiated and it would be interesting to monitor how miRNA profiles develop upon differentiation to a more mature neuron-like state. The composition of miRNAs for each cell line is not fixed, and could be adjusted experimentally, for example by differentiation or introduction or deletion of miRNAs or miRNA maturation components. It would be interesting to observe if such manipulations (e.g., of miR-124, miR-134) resulted in acquisition of neuronal projection features such as axons or dendrites.

Finally, our study did not find a large effect of species, with most miRNA profiles quite conserved between human and mouse lines. This supports the relevance of the mouse lines for miRNA functions in human diseases. There are, however, human-specific miRNAs, many of which are expressed uniquely in the brain [56], which should be considered according to the research question. Regardless, these findings suggest that the cellular function of interest should be the primary decision point in choosing the cellular line rather than the species origin.

There are some limitations in the present study. First, we used a qPCR-based system to profile miRNAs. The OpenArray platform has a pre-selected number of miRNAs such that we have not captured the full repertoire of miRNAs. We will also have missed isomiRs and editing, which are important for miRNA function [57]. Second, we did not include human primary (e.g. iPSC) neurons which would provide a useful additional comparator. Third, our study only evaluated the basal levels of miRNA expression and cannot be used to infer how the cell line miRNA profiles may respond to stressors. And last, in the current study, we have compared immortalized cell lines to mature hippocampal neurons, and not to neural progenitor cells. The limitation here, is the difference in maturation state and we cannot be certain whether the cell lines are more similar to progenitors or mature neurons. This could be explored in future experiments, comparing progenitors to varying states of maturation using primary hippocampal neurons.

In conclusion, we used a low-input profiling platform to screen miRNAs in three common cell lines and compared findings to primary neurons. We found overall a high degree of correlation indicating these lines are suitable 'models' for the identification of miRNA-related processes and their target genes. However, specific miRNA levels varied between lines with some miRNA absent while each line varied in terms of aligning to primary neurons. The results provide a catalogue of miRNA expression which researchers can use to select the most appropriate cell type for basic and applied miRNA research, including for studies on epilepsy.

## Supporting information

**S1 Fig. Venn diagrams represent the overlap between the number of miRNAs expressed in our experiment and by chance after 1000 bootstrapping iterations.**
(PDF)

**S2 Fig. Z-score Heat Map of the 98 miRNAs expressed in all four cell types.**
(PDF)

**S3 Fig. Dot plot illustrating the enriched biological pathways of the miRNAs commonly expressed in all cell lines.**
(PDF)

**S4 Fig. Dot plot illustrating the enriched biological pathways of the microRNAs exclusively expressed in each cell line.**
(PDF)

**S1 Methods. Extended methods used on the generation of our supplementary methods.**
(PDF)

## Acknowledgments

We would like to thank Carsten Culmsee (Philipps-Universität Marburg) for the HT22 cell line and Jochen Prehn (Royal College of Surgeons in Ireland) for the N2A and SH-SY5Y cell lines. We also thank James Mills for help with the analysis of the miRNA data from human brain tissue.

## Author contributions

**Conceptualization:** Eva Maria Jimenez Mateos, David C Henshall.

**Formal analysis:** Ronan Murphy, Javier Villegas-Salmeron.

**Investigation:** Ronan Murphy, Amaya Sanz-Rodriguez.

**Methodology:** Tobias Engel, Catherine Mooney.

**Writing – original draft:** Eva Maria Jimenez Mateos, David C Henshall.

**Writing – review & editing:** Eva Maria Jimenez Mateos, David C Henshall.

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
