## [Decision Letter · Decision Letter 0]

10 Jul 2025

Dear Dr. Jimenez Mateos,

Thank you for submitting your manuscript to PLOS ONE. After careful consideration, we feel that it has merit but does not fully meet PLOS ONE’s publication criteria as it currently stands. Therefore, we invite you to submit a revised version of the manuscript that addresses the points raised during the review process.

We look forward to receiving your revised manuscript.

Kind regards,

Faramarz Dehghani

Academic Editor

PLOS ONE

Journal Requirements:

This publication has emanated from research conducted with the financial support of the European Union's ‘Seventh Framework’ Programme (FP7) under Grant Agreement no. 602130 to DCH. Additional support is from Health Research Board under Grant number ILP-POR-2022-029, Wellcome Trust-IISF call under Grant number Ph2IISF-16585  to EMJM, and Taighde Éireann – Research Ireland, under Grant number 17/CDA/4708 (TE), 22/FFP-P/11333 (EMJM) and 21/RC/10294_P2 at FutureNeuro Research Ireland Centre for Translational Brain Science (DCH).

We would like to thank Carsten Culmsee (Philipps-Universität Marburg) for 349 the HT22 cell line and Jochen Prehn (Royal College of Surgeons in Ireland) for the N2A and SH350 SY5Y cell lines. We also thank James Mills for help with analysis of the miRNA data from human 351 brain tissue. This publication has emanated from research conducted with the financial support of the 352 European Union's ‘Seventh Framework’ Programme (FP7) under Grant Agreement no. 602130. 353 Additional support is from Health Research Board under Grant number ILP-POR-2022-029, 354 Wellcome Trust-IISF call under Grant number Ph2IISF-16585, and Taighde Éireann – Research 355 Ireland, under Grant number 17/CDA/4708, 22/FFP-P/11333 and 21/RC/10294_P2 at FutureNeuro 356 Research Ireland Centre for Translational Brain Science.

This publication has emanated from research conducted with the financial support of the European Union's ‘Seventh Framework’ Programme (FP7) under Grant Agreement no. 602130 to DCH. Additional support is from Health Research Board under Grant number ILP-POR-2022-029, Wellcome Trust-IISF call under Grant number Ph2IISF-16585  to EMJM, and Taighde Éireann – Research Ireland, under Grant number 17/CDA/4708 (TE), 22/FFP-P/11333 (EMJM) and 21/RC/10294_P2 at FutureNeuro Research Ireland Centre for Translational Brain Science (DCH).

5. We noted in your submission details that a portion of your manuscript may have been presented or published elsewhere. [We compared our microRNA profile with previously data published, this has been acknowledged on the manuscript on the respective section. Data of mouse microRNA profile was published in Jimenez-Mateos et al., 2015 and human data was published previously in Mills et al., 2020.] Please clarify whether this [publication] was peer-reviewed and formally published. If this work was previously peer-reviewed and published, in the cover letter please provide the reason that this work does not constitute dual publication and should be included in the current manuscript.

7. Please amend your list of authors on the manuscript to ensure that each author is linked to an affiliation. Authors’ affiliations should reflect the institution where the work was done (if authors moved subsequently, you can also list the new affiliation stating “current affiliation:….” as necessary).

Reviewers' comments:

Reviewer's Responses to Questions

**Comments to the Author**

1. Is the manuscript technically sound, and do the data support the conclusions?

Reviewer #1: Yes

Reviewer #2: Yes

2. Has the statistical analysis been performed appropriately and rigorously?

Reviewer #1: Yes

Reviewer #2: Yes

3. Have the authors made all data underlying the findings in their manuscript fully available?

Reviewer #1: No

Reviewer #2: No

4. Is the manuscript presented in an intelligible fashion and written in standard English?

Reviewer #1: Yes

Reviewer #2: Yes

Reviewer #1: The manuscript by Murphy et al., entitled “Shared and distinct microRNA profiles between HT22, N2A and SH-SY5Y cell lines and primary mouse hippocampal neurons”, is aiming to compare microRNA profiles between common used neuronal hippocampal cell lines and primary hippocampal neurons.

The manuscript is interesting for the field; however, some points need to be addressed before the publication.

The detailed review points can be found below:

1. Please attach the raw data of the miRNA analysis.

2. Why hippocampus and a hippocampal cell line were used as a control, since only HT22 cells originate from the hippocampus?

3. A bioethical statement for laboratory animal use is missing

4. Were the cells authenticated and tested for mycoplasma?

5. Authors aimed to compare the expression profile of cells based on similarity using Pearson correlation analysis. This type of analysis is only a poor descriptor of similarity. I recommend using a different metric for this.

6. Authors stated on several occasions percentages of overlapping miRNAs expressed in their cell line. While these numbers are interesting, they should be considered in comparison to the number expected by chance for the given number of expressed miRNA.

7. Authors talked about the function of selected miRNA in their manuscript. I highly recommend the authors to do a functional association analysis (data base analysis) of all miRNA that are expressed jointly in all cell lines/primary cells and all those that differ between the cell lines. As authors propose, one main goal of the manuscript is to characterize these cell lines for potential further use cases it is of importance to narrow down the functional implications of differential and similar miRNA expression as much as possible.

8. Figure 2D: Please use a different way to illustrate the data as it is hard to follow the individual expression of those miRNAs, e.g. as a heatmap of z-scores. Please provide the raw data for this figure as a supplemental table.

9. Fig 4 is blurred. Please exchange it

10. Table 1 is twice in the manuscript

11. How was the hippocampal tissue isolated, which mouse strain and how old were the mice?

12. In Fig 1 legend several typos “tisuue” and “hypocampal” are present

Reviewer #2: This study investigates the miRNA expression profiles of three commonly used neuronal cell lines—HT22, N2A, and SH-SY5Y—by comparing them to primary mouse hippocampal neurons. The study found a high correlation in miRNA profiles (especially for HT22 line), supporting their suitability as models for studying miRNA-related processes and target genes. However, specific miRNA levels differed between lines and primary neurons, offering a useful catalogue of miRNA expression to help researchers choose the most appropriate cell model for miRNA studies.

Overall, the methodological approach adopted by the authors is appropriate and well-executed, with relevant questions being addressed in relation to the biological problem at hand. However, I have three major comments and a minor one.

The minor comment concerns the use of tissue from epileptic patients. These samples are neither adequately referenced nor described in the Results and Discussion sections. Simply stating “human hippocampus” is not sufficient.

The three major comments are:

1. I could not find any supplementary material containing the primary data used in this study, which is essential to verify the analyses and the figures presented. Without access to these data, the results are not reproducible and cannot be readily reused by other researchers.

2. The authors mention the lack of miRNA profiling in differentiated cells of the tested lines as a limitation. I believe this is a critical point for the study’s objectives. While I understand the technical and resource demands of such an experiment (and that it may be part of future work), a minimum step should be to compare neural stem cells with primary neurons. For this purpose, RNA could be extracted directly from mouse cortices at E11/E12 (the peak of neural stem cell proliferation), or alternatively cultured until E14 under proliferative conditions.

This comparison would be valuable for two reasons:

a) To determine whether there is a significant shift in miRNA profiles between neural progenitors and differentiated/mature neurons. It is also possible that the profiles remain largely similar.

b) Additionally, it would help in fitting the profiles obtained from the various cell lines along a continuum from progenitors to mature neurons (especially if a significant difference is observed in point a).

Two possible scenarios may emerge:

• If no significant difference is observed between progenitors and mature neurons, the conclusions of the study would shift toward identifying which lines most closely resemble general neural cells, rather than neurons specifically.

• If a clear difference exists, it would then be possible to classify each cell line as being more similar to either progenitors or mature neurons.

3. Functional analyses (e.g. miRNet 2.0 computational tool) should be performed on the differentially expressed miRNAs across the various comparisons shown in Figures 4 and 5. The current manuscript only highlights a few selected miRNAs, despite many others showing differential expression. On what statistical or biological criteria were these specific miRNAs chosen? This selection needs to be clarified and justified.

**Do you want your identity to be public for this peer review?** For information about this choice, including consent withdrawal, please see our Privacy Policy

Reviewer #1: No

Reviewer #2: No

---

## [Author Response · Author response to Decision Letter 1]

11 Sep 2025

We would like to thank the reviewers for the comments that have helped to improve our manuscript. As requested by both reviewers, we have uploaded the raw data in the Supplementary Information as an Excel file. Please find below a detailed point-by-point discussion of the concerns and suggestions raised during the review process. Changes to the revised manuscript are highlighted in the text.

Reviewer 1: We thank the reviewer for their positive assessment of our manuscript. Please find below the response to each comment point-by-point:

1. Please attach the raw data of the miRNA analysis.

Answer: The raw data has been made available in the Supplementary Information as an Excel file. This information has been updated in the manuscript under the Methods section, Analysis of Open Array subsection, page 8, lines 166-167.

2. Why hippocampus and a hippocampal cell line were used as a control, since only HT22 cells originate from the hippocampus?

Answer: We thank the reviewer for raising this point. As the reviewer correctly points out, HT22 is the only cell line with a hippocampal origin. The rationale for the inclusion of two additional [non-hippocampal] cell lines was two-fold. First, to provide context for the microRNA profile of the HT22 cell lines by comparing them to other immortalised cell lines (vs primary neurons). This enables the reader to better appreciate what signatures might reflect tissue of origin vs immortalisation. Second, the widespread use of the other two cell lines in neuroscience research makes their inclusion an obvious strength and point of general interest. Despite not having a hippocampal tissue origin, we nevertheless identified microRNA profiles that resembled primary neurons. This may provide assurance to researchers using these non-hippocampal lines that their findings will bear relevance to primary hippocampal neurons (in the event they choose not to switch to HT22 for other reasons).

We have clarified the value of including the other two cell lines in our revised manuscript in the Discussion section page 14, lines 302-37: “Importantly, the current study compared hippocampal primary neurons not only to the HT22 line which are of hippocampal origin, but also neuroblastoma cell lines N2A and SH-SY5Y. This adds important value to the present findings in two ways. First, the inclusion enables the identification of microRNA signatures that reflect tissue origin versus immortalisation. Second, the inclusion of these cell lines gives the fingerprint of the similarities between both neuroblastoma cell lines and primary hippocampal neurons”.

3. A bioethical statement for laboratory animal use is missing

Answer: All procedures were performed in accordance with the guidelines of the European Communities Council Directives (86/609/EU and 2010/63/EU) and were reviewed and approved by the Research Ethics Committee of the Royal College of Surgeons in Ireland (REC #765) under license from the Department of Health, Dublin, Ireland (B100/4524). We have updated this information on the Methods section, primary cell culture subsection, page 6, lines 111-114.

4. Were the cells authenticated and tested for mycoplasma?

Answer: All cell lines were tested for mycoplasmas every 6 months; cell lines used in those experiments tested negative for the presence of mycoplasmas. We have updated the cell culture subsection in the methods to contain this information as: “Cell lines were tested for mycoplasmas before experiments started. Results were negative for the three immortalized cell lines” (Methods section, cultured cell line subsection, page 7, lines 136-137)

5. Authors aimed to compare the expression profile of cells based on similarity using Pearson correlation analysis. This type of analysis is only a poor descriptor of similarity. I recommend using a different metric for this.

Answer: To validate the Pearson correlation analysis, we have used the Mean Squared Error as a measure of similarity. Using this approach our HT22 cell have a score of 8.515, N2A cells have a score of 13.444 and SH-SY5Y cells have a score of 8.443. This data strongly aligns with our previous analysis. This information has been added to the text in the methods section page 8, lines 158-159 (Correlation was calculated using the Pearson correlation coefficient and the Mean Squared Error to evaluate similarity), results section page 12, lines 255-257, and in the Figure 4 legend (Correlations between the expression levels of the individual microRNAs between the primary hippocampal neurons and A) HT22 (r=0.700, MSE=8.515); B) N2A (r=0.564, MSE=13.444); and C) SH-SY5Y (r=0.721, MSE=8.443).

6. Authors stated on several occasions percentages of overlapping miRNAs expressed in their cell line. While these numbers are interesting, they should be considered in comparison to the number expected by chance for the given number of expressed miRNA.

Answer: Assuming that all possibilities are equally likely, we have a total of 16 scenarios, A single microRNA can be expressed in all of the groups, none of the groups, exclusively expressed in one group or expressed in a selected number of groups (2 or 3 of them). Taking into consideration these scenarios and that these are equally possible, the chance of a microRNA expressed in all groups is 6.25%. This strongly supports that our profile is not due to a random chance.

7. Authors talked about the function of selected miRNA in their manuscript. I highly recommend the authors to do a functional association analysis (data base analysis) of all miRNA that are expressed jointly in all cell lines/primary cells and all those that differ between the cell lines. As authors propose, one main goal of the manuscript is to characterize these cell lines for potential further use cases it is of importance to narrow down the functional implications of differential and similar miRNA expression as much as possible.

Answer: Thank you for the comment. We have generated two new Figures (Supplementary Figure 2 and 3) to represent the enriched pathways regulated by the microRNAs expressed in all cell lines and primary hippocampal neurons (Supp. Figure 2), and unique to each cell type (Supp. Figure 3). Regarding the first figure, we have also added the enriched biological pathways regulated by the commonly expressed microRNAs between the primary hippocampal neurons and each of the immortalized cell lines as requested by Reviewer 2, Point 3.

We have included this information in the Results section pages 12-13, lines 272-283: “ Then, we evaluated the biological pathways enriched in all cell lines combined, primary cell cultures and/or each cell line. Axon guidance, cell-to-cell communication, and transport of molecules were the most significantly regulated within the 98 common microRNAs expressed in all the cell types (Supp. Figure 2). Metabolism was the main pathway regulated by microRNAs expressed in primary hippocampal neurons and HT22. Axon guidance and developmental biology was found between primary hippocampal neurons and N2A. And at last, RNA regulation and post-transcriptional regulation, between primary hippocampal neurons and SH-SY5Y cells (Supp Figure 2). Finally, we evaluated the microRNAs-regulated pathways by the specific microRNAs expressed exclusively in each cell line. This revealed that pathways involved in Calcium regulation and CREB transcription factor signalling activation are only seen in the primary cell culture, while mTOR and S6K signalling pathways are only seen in HT22 cell line, extracellular matrix regulation is observed only in N2A cell lines, and, finally, in SH-SY5Y cell lines has pathways related to cytokine regulation and innate immune system (Supp. Figure. 3).”

Methodology for this section has been included in the supplementary information, under the subheading supplementary methods.

8. Figure 2D: Please use a different way to illustrate the data as it is hard to follow the individual expression of those miRNAs, e.g. as a heatmap of z-scores. Please provide the raw data for this figure as a supplemental table.

Answer: Thank you for the suggestion. We propose to retain the original Figure 2 since a key value is that it shows relative abundance of each microRNA in the different cell lines. However, we have generated a z-score heat map to complement the data in Figure 2. This has been included as supplementary information (Suppl Figure 1) and in the text page 10, line 219.

9. Fig 4 is blurred. Please exchange it

Answer: Thank you for the comment. We have uploaded a higher-quality image of Figure 4.

10. Table 1 is twice in the manuscript

Answer: Thank you for the comment. We have removed one of the copies of Table 1 from the manuscript.

11. How was the hippocampal tissue isolated, which mouse strain and how old were the mice?

Answer: For the primary hippocampal cell cultures, C57BL/6J pregnant females were obtained from the Biomedical Research Facilities (RCSI). Embryonic day 18 (E18) old pups were used for these experiments. Hippocampi were isolated and tissue dissociated using the papain protocol as previously described by Jimenez-Mateos et al., 2012. This information has been clarified in the Methods section, primary cell culture subsection, page 6, lines 116-124.

12. In Fig 1 legend several typos “tisuue” and “hypocampal” are present

Answer: Thank you for the comment. Typos have been corrected in the Figure legend, see page 19 lines 413 and 414, respectively.

Reviewer 2: We thank the reviewer for their comments and suggestions. Please find below the response to each comment point-by-point:

1. The minor comment concerns the use of tissue from epileptic patients. These samples are neither adequately referenced nor described in the Results and Discussion sections. Simply stating “human hippocampus” is not sufficient.

Answer: The data used in the current manuscript have been previously published in Mills et al., 2019 (Ref. 46 in manuscript). From this original manuscript, we used data from 3 hippocampal post-mortem control tissues after obtaining permission from the authors (see acknowledgement section). Control material was obtained during autopsy resection, and a history of no seizures or other neurological diseases was confirmed.

To clarify this, we have revised the manuscript and have added the following information in the Methods section, Human data and RNA-Seq data analysis subsection page 8, lines 170-172: “RNA-seq raw data was obtained from previously available data [46]. Hippocampal post-mortem tissue was obtained during autopsy resection and a history of no-seizures or other neurological diseases was confirmed as described in the original publication [46].”

The three major comments are:

1. I could not find any supplementary material containing the primary data used in this study, which is essential to verify the analyses and the figures presented. Without access to these data, the results are not reproducible and cannot be readily reused by other researchers.

Answer: The raw data has been made available at Supplementary Information as an Excel file. This information has been updated in the manuscript under the Methods section, Analysis of Open Array subsection, page 8, lines 166-167.

2. The authors mention the lack of miRNA profiling in differentiated cells of the tested lines as a limitation. I believe this is a critical point for the study’s objectives. While I understand the technical and resource demands of such an experiment (and that it may be part of future work), a minimum step should be to compare neural stem cells with primary neurons. For this purpose, RNA could be extracted directly from mouse cortices at E11/E12 (the peak of neural stem cell proliferation), or alternatively cultured until E14 under proliferative conditions.

This comparison would be valuable for two reasons:

a) To determine whether there is a significant shift in miRNA profiles between neural progenitors and differentiated/mature neurons. It is also possible that the profiles remain largely similar.

b) Additionally, it would help in fitting the profiles obtained from the various cell lines along a continuum from progenitors to mature neurons (especially if a significant difference is observed in point a).

Two possible scenarios may emerge:

• If no significant difference is observed between progenitors and mature neurons, the conclusions of the study would shift toward identifying which lines most closely resemble general neural cells, rather than neurons specifically.

• If a clear difference exists, it would then be possible to classify each cell line as being more similar to either progenitors or mature neurons.

Answer: We thank the reviewer for this valuable point and suggesting to analyse neural progenitors as an additional comparator. Unfortunately, the PCR-based Open Array platform used to generate the profiles in the original study has notable batch-to-batch effects, which would make such a direct comparison problematic. We consider the use of previously available data; however, all open access data found by the authors on neural progenitor cells were not suitable for this study, as they were transgenic mice or treated with vehicles, such as ethanol or DMSO. Nevertheless, we consider this a good point, and we have added the following paragraph in the discussion (page 16, lines 365-369): “And last, in the current study, we have compared immortalized cell lines to mature hippocampal neurons, and not to neural progenitor cells. The limitation here, is the difference in maturation state and we cannot be certain whether the cell lines are more similar to progenitors or mature neurons. This could be explored in future experiments, comparing progenitors to varying states of maturation using primary hippocampal neurons”.

3. Functional analyses (e.g. miRNet 2.0 computational tool) should be performed on the differentially expressed miRNAs across the various comparisons shown in Figures 4 and 5. The current manuscript only highlights a few selected miRNAs, despite many others showing differential expression. On what statistical or biological criteria were these specific miRNAs chosen? This selection needs to be clarified and justified.

Answer: Thank you for the comment. We have generated a new Figure (Supplementary Figure 2) to represent the enriched biological pathways regulated by the commonly expressed microRNAs between the primary hippocampal neurons and the cell lines (Figure 4) and microRNA commonly expressed in all the cell lines and primary hippocampal neurons as requested by Reviewer 1 (see Reviewer 1, point 6).

We have included this information on the Results section pages 12-13, lines 272-283: “ Then, we evaluated the biological pathways enriched in all cell lines combined, primary cell cultures and/or each cell line. Axon guidance, cell-to-cell communication, and transport of molecules were the most significantly regulated within the 98 common microRNAs expressed in all the cell types (Supp. Figure 2). Metabolism was the main pathway regulated by microRNAs expressed in primary hippocampal neurons and HT22. Axon guidance and developmental biology was found between primary hippocampal neurons and N2A. And at last, RNA regulation and post-transcriptional regulation, between primary hippocampal neurons and SH-SY5Y cells (Supp Figure 2). Finally, we evaluated the microRNAs-regulated pathways by the specific microRNAs expressed exclusively in each cell line. This revealed that pathways involved in Calcium regulation and CREB transcription factor signalling activation are only seen in the primary cell culture, while mTOR and S6K signalling pathways are only seen in HT22 cell line, extracellular matrix regulation is observed only in N2A cell lines, and, finally, in SH-SY5Y cell lines has pathways related to cytokine regulation and innate immune system (Supp. Figure. 3).”

On figure 5, we decided to show a representation of microRNAs to emphasize the similarities between primary hippocampal neurons and the cell lines. For this, we focused on microRNAs linked to neuronal function. This has been added for clarification on the Results section page 12 lines 285-286:” particularly those microRNAs previously linked to neuronal function.”

Methodology for this section has been included in the supp

---

## [Decision Letter · Decision Letter 1]

8 Oct 2025

Dear Dr. Jimenez Mateos,

We look forward to receiving your revised manuscript.

Kind regards,

Faramarz Dehghani

Academic Editor

PLOS ONE

Journal Requirements:

Reviewers' comments:

Reviewer's Responses to Questions

**Comments to the Author**

Reviewer #1: (No Response)

Reviewer #2: All comments have been addressed

2. Is the manuscript technically sound, and do the data support the conclusions?

Reviewer #1: Yes

Reviewer #2: Yes

3. Has the statistical analysis been performed appropriately and rigorously?

Reviewer #1: No

Reviewer #2: Yes

4. Have the authors made all data underlying the findings in their manuscript fully available?

Reviewer #1: Yes

Reviewer #2: Yes

5. Is the manuscript presented in an intelligible fashion and written in standard English?

Reviewer #1: Yes

Reviewer #2: Yes

Reviewer #1: The manuscript by Murphy et al., entitled “Shared and distinct microRNA profiles between HT22, N2A and SH-SY5Y cell lines and primary mouse hippocampal neurons”, is aiming to compare microRNA profiles between common used neuronal hippocampal cell lines and primary hippocampal neurons.

The manuscript is interesting for the field; and this version is significantly improved, almost all of the critic points were addressed.

However, one major point need to be still addressed.

Point 6:

When calculating the possibility for the overlap to be caused by chance, the approach of the authors is sub-par, as it does not include the proportion of expressed/differently expressed (depending on the context) miRNAs relative to the total amount of miRNAs tested for. As a thought experiment to illustrate this idea imagine a tumor cell line expressing all tested miRNAs and comparing them to any cell line. Thus, this analysis would deem them highly similar and would fail to identify the result as being caused by chance. Consequently, another estimator to account for randomness is needed. A typical approach could be based on bootstrapping. E.g. authors could calculate the percentage of miRNAs expressed or over-expressed (depending on the context) in one of the samples to be compared, and than randomly resample the miRNAs expressed/over-expressed to create a virtual new data-set and now check the overlap between the random set and the data set to be compared to . This corresponds to one bootstrapping iteration. This procedure needs to be repeated at least 1000 times to obtain 1000 values for the percantage of overlap. The mean value of these 1000 percentages would be a descent estimate for the probability for the amount of overlap expected due to chance alone, independent on biological similarity.

Reviewer #2: I thanks the Authors that have adequately addressed my previous comments.

I have only two minor comments/suggestions at:

- line 56, modify "Argonuate" in "Argonaute"

- line 170, I suggest to modify in "RNA-seq raw data of hippocampus of control human donor was obtained from previously available data []. Hippocampal post-mortem tissue ..."

**Do you want your identity to be public for this peer review?** For information about this choice, including consent withdrawal, please see our Privacy Policy

Reviewer #1: No

Reviewer #2: No

---

## [Author Response · Author response to Decision Letter 2]

3 Nov 2025

We would like to thank the reviewer for the further comments that have helped to improve our manuscript.

Reviewer 1: We thank the reviewer for their positive assessment of our manuscript. Please find below the response to each comment point-by-point:

1. Point 6: When calculating the possibility for the overlap to be caused by chance, the approach of the authors is sub-par, as it does not include the proportion of expressed/differently expressed (depending on the context) miRNAs relative to the total amount of miRNAs tested for. As a thought experiment to illustrate this idea imagine a tumor cell line expressing all tested miRNAs and comparing them to any cell line. Thus, this analysis would deem them highly similar and would fail to identify the result as being caused by chance. Consequently, another estimator to account for randomness is needed. A typical approach could be based on bootstrapping. E.g. authors could calculate the percentage of miRNAs expressed or over-expressed (depending on the context) in one of the samples to be compared, and than randomly resample the miRNAs expressed/over-expressed to create a virtual new data-set and now check the overlap between the random set and the data set to be compared to . This corresponds to one bootstrapping iteration. This procedure needs to be repeated at least 1000 times to obtain 1000 values for the percantage of overlap. The mean value of these 1000 percentages would be a descent estimate for the probability for the amount of overlap expected due to chance alone, independent on biological similarity.

Answer: Thank you for the comment. We have now run a Bootstrapping Re-sampling analysis to account for random distribution. We have generated a new Supplementary Figure 1. Supp. Figure 1 shows side-by-side the distribution of microRNAs in our experiment and compares it to the Resampling set. We have update the information on page 8 line 168-172: “Further analysis was carried out to evaluate if the distribution of the number of microRNAs expressed in each condition is due to chance using the Bootstrapping resampling method. Bootstrapping iterations were performed using random binomial resampling with probability equal to the percentage of microRNAs expressed by each cell type in the original data (see Supp. Figure 1 and methods)”. On page 10 lines 221-225:”To determine whether the observed overlap in miRs between the different cell lines was due to random chance, we performed 1000 bootstrapping iterations and compared the median results to our original data (Supp. Figure 1 and methods). This revealed that the resampled distribution was significantly different from the distribution in our profile (Supp. Figure 1), and thus the overlap in miR expression across the cell lines is likely due to biological similarity rather than random chance.”. And we have added the new methods to the Supplementary data section:

“ Bootstrapping analysis for the evaluation of microRNA distribution by chance.

To explore whether the distribution of microRNAs in the different cell lines might be simply random/chance, we used a bootstrapping approach to randomly determine the microRNA profile in each condition. To achieve this, we took the 310 microRNAs expressed by any of the four cell types and generated a random binomial distribution for each cell type using rbinom in R, with probability equal to the percentage of microRNAs expressed by that cell type in the original data. We performed 1000 iterations and compared the number of microRNAs in each condition (Supp. Figure 1). Supp. Figure 5 shows the histogram for the number of microRNAs expressed by all four cell types in each bootstrapping iteration. The median number of commonly expressed miRNAs was 43 (95% CI = 32, 55). In our study, we observed 98 commonly expressed microRNAs, far above the upper range of the confidence interval. This suggests that the number of commonly expressed microRNAs is due to biological similarity rather than random chance. This approach was used for each of the conditions; the median value of each interaction was used to generate Supp. Figure 1.”

Reviewer 2: We thanks Reviewer 2 for the minor comments/suggestions. See our response below point by point :

1. Line 56, modify "Argonuate" in "Argonaute"

Answer: We have corrected this typo.

2. Line 170, I suggest to modify in "RNA-seq raw data of hippocampus of control human donor was obtained from previously available data []. Hippocampal post-mortem tissue ..."

Answer: The line has been corrected following the reviewer’s recommendation (Line 175).

---

## [Decision Letter · Decision Letter 2]

6 Nov 2025

Shared and distinct microRNA profiles between HT22, N2A and SH-SY5Y cell lines and primary mouse hippocampal neurons

PONE-D-25-28988R2

Dear Dr. Jimenez Mateos,

We’re pleased to inform you that your manuscript has been judged scientifically suitable for publication and will be formally accepted for publication once it meets all outstanding technical requirements.

Kind regards,

Faramarz Dehghani

Academic Editor

PLOS ONE

Additional Editor Comments (optional):

Reviewers' comments:

Reviewer's Responses to Questions

**Comments to the Author**

Reviewer #1: All comments have been addressed

2. Is the manuscript technically sound, and do the data support the conclusions?

Reviewer #1: Yes

3. Has the statistical analysis been performed appropriately and rigorously?

Reviewer #1: Yes

4. Have the authors made all data underlying the findings in their manuscript fully available?

Reviewer #1: Yes

5. Is the manuscript presented in an intelligible fashion and written in standard English?

Reviewer #1: Yes

Reviewer #1: (No Response)

**Do you want your identity to be public for this peer review?** For information about this choice, including consent withdrawal, please see our Privacy Policy

Reviewer #1: No

---

## [Editor Report · Acceptance letter]

PONE-D-25-28988R2

PLOS ONE

Dear Dr. Jimenez Mateos,

I'm pleased to inform you that your manuscript has been deemed suitable for publication in PLOS ONE. Congratulations! Your manuscript is now being handed over to our production team.

Kind regards,

on behalf of

Dr. Faramarz Dehghani

Academic Editor

PLOS ONE